# Changes in Striatal Medium Spiny Neuron Morphology Resulting from Dopamine Depletion Are Reversible

**DOI:** 10.3390/cells9112441

**Published:** 2020-11-09

**Authors:** Victoria Sofie Witzig, Daniel Komnig, Björn H. Falkenburger

**Affiliations:** 1Department of Neurology, RWTH Aachen University, 52074 Aachen, Germany; victoria.witzig@rwth-aachen.de (V.S.W.); daniel.komnig@rwth-aachen.de (D.K.); 2JARA-Institute Molecular Neuroscience and Neuroimaging, Forschungszentrum Jülich GmbH and RWTH Aachen University, 52074 Aachen, Germany; 3Department of Neurology, Technische Universität Dresden, 01307 Dresden, Germany; 4Deutsches Zentrum für Neurodegenerative Erkrankungen, 01307 Dresden, Germany

**Keywords:** spiny projection neurons, striatum, spine density, dendrite morphology

## Abstract

The classical motor symptoms of Parkinson’s disease (PD) are caused by degeneration of dopaminergic neurons in the substantia nigra, which is followed by secondary dendritic pruning and spine loss at striatal medium spiny neurons (MSN). We hypothesize that these morphological changes at MSN underlie at least in part long-term motor complications in PD patients. In order to define the potential benefits and limitations of dopamine substitution, we tested in a mouse model whether dendritic pruning and spine loss can be reversible when dopaminergic axon terminals regenerate. In order to induce degeneration of nigrostriatal dopaminergic neurons we used the toxicity of 1-methyl-4-phenyl-1,2,3,6-tetrahydropyridine (MPTP) in C57BL/6J mice; 30 mg/kg MPTP was applied i.p. on five consecutive days. In order to assess the consequences of dopamine depletion, mice were analyzed 21 days after the last injection. In order to test reversibility of MSN changes we exploited the property of this model that striatal axon terminals regenerate by sprouting within 90 days and analyzed a second cohort 90 days after MPTP. Degeneration of dopaminergic neurons was confirmed by counting TH-positive neurons in the substantia nigra and by analyzing striatal catecholamines. Striatal catecholamine recovered 90 days after MPTP. MSN morphology was visualized by Golgi staining and quantified as total dendritic length, number of dendritic branch points, and density of dendritic spines. All morphological parameters of striatal MSN were reduced 21 days after MPTP. Statistical analysis indicated that dendritic pruning and the reduction of spine density represent two distinct responses to dopamine depletion. Ninety days after MPTP, all morphological changes recovered. Our findings demonstrate that morphological changes in striatal MSN resulting from dopamine depletion are reversible. They suggest that under optimal conditions, symptomatic dopaminergic therapy might be able to prevent maladaptive plasticity and long-term motor complications in PD patients.

## 1. Introduction

The classic motor symptoms of Parkinson disease (PD) are tremor, rigidity and bradykinesia. They result from the degeneration of dopaminergic neurons in the *substantia nigra* (SN) and the resulting dopamine deficiency in the striatum. Accordingly, the cardinal PD motor symptoms are alleviated by the dopamine precursor levodopa and by dopamine receptor agonists [1,2]. Motor symptoms remain responsive to these dopaminergic medications throughout the course of PD. In advanced PD, however, symptom control is hampered by the fact that a single dose of medication triggers dyskinesia more easily and lasts for a shorter period of time. Presynaptic mechanisms can explain some features of these motor fluctuations but not their induction by dopamine receptor agonists [3,4].

Postsynaptic mechanisms include the fact that neurons generally do not remain passive in the face of major changes in synaptic inputs. Homeostatic plasticity allows neuronal networks to maintain a stable rate of activity when synapses change [5]. This has been studied mainly during development and learning [5,6], but also in the course of neurological and psychiatric diseases [7,8,9,10]. In humans and animal models, chronic dopaminergic depletion and substitution can induce plastic changes that differ from acute effects and that are potentially mediated by homeostatic plasticity mechanisms. These effects include the “long duration response” (LDR) to dopaminergic substitution, levodopa-induced dyskinesias (LID) and tardive dyskinesias [11,12].

The consequences of chronic dopaminergic denervation have been primarily assessed in striatal medium spiny neurons (MSN, also called spiny projection neurons) because they are the main postsynaptic neurons for dopaminergic axon terminals. Effects differ between MSN expressing excitatory D1 dopamine receptors (D1-MSN) and MSN expressing inhibitory D2 dopamine receptors (D2-MSN). For instance, D2-MSN are disinhibited by dopamine depletion. The homeostatic response in the face of chronic dopamine depletion includes a decrease in electrical excitability and a reduced number of glutamatergic synapses [8]. A lower density of dendritic spines and a pruned dendritic tree was also observed in further animal models [9,13,14,15,16,17,18] and in striatal MSN of PD patients [19,20].

In order to guide dopaminergic substitution in PD patients, it is important to know whether these changes in MSN morphology can be reversible, and what governs reversibility. In previous work, reversibility was tested by administration of levodopa in animals with degeneration of dopaminergic neurons induced by injection of 6-hydroxydopamine (6-OHDA) or in aphakia (Pitx3-/-) mice. The reduction in spine density was found to be reversible in D2-MSN but not in in D1-MSN [21,22,23]. Dendritic pruning, in contrast, was not reversible [8]. This indicates that reversibility requirements may differ between D1 and D2 MSN and between spine density and TDL. It furthermore raises the question of whether a different regimen of dopaminergic substitution would allow a reversibility of dendritic pruning.

In order to investigate reversibility of MSN morphology under optimal conditions, we used the 1-methyl-4-phenyl-1,2,3,6-tetrahydropyridine (MPTP) model of PD. In this model, dopaminergic neurons in the substantia nigra and their axon terminals in the striatum degenerate after MPTP administration. Nigrostriatal axon terminals in the striatum recover 3 months after MPTP due to axonal sprouting [24], which does not happen in the 6-OHDA model. Because dopamine is secreted from the recovering nigrostriatal axon terminals, there is no need to optimize the dosing and timing of levodopa administration to revert morphological changes in the postsynaptic MSN.

In this study we thus confirmed that MPTP-induced dopamine depletion in the striatum alters MSN morphology by using Golgi staining at the peak of dopaminergic denervation—21 days after MPTP administration. We then determined which of these changes are reversible at 90 days after MPTP when dopaminergic axon terminals have regenerated.

## 2. Methods

### 2.1. MPTP Mice

C57BL/6J mice (RRID: IMSR_JAX:000664) were housed and handled in a pathogen-free animal facility at 20–24 °C with a 12 h light/dark cycle, food and water ad libitum, in accordance with guidelines of the Federation for European Laboratory Animal Science Associations (FELASA). Procedures were approved by the local authorities (Landesamt für Natur, Umwelt und Verbraucherschutz Nordrhein-Westfalen, 84-02.04.2014.A171). Mice were 10–12 weeks of age at the start of the experiment. They received either MPTP hydrochloride in 0.9% saline or saline alone. MPTP was administered at a dose of 30 mg free base per kg body weight i.p. at 24 h intervals over 5 consecutive days. MPTP handling and safety measures were in accordance with published guidelines [25]. One group of animals was sacrificed 21 days after the last MPTP injection by cervical dislocation under deep isoflurane anesthesia. A second group of animals was sacrificed 90 days after the last MPTP injection. Mice were monitored daily for physical condition and weight loss. One animal died on day 5, all others that started the experiment survived with <20% weight loss and were included into the analyses.

After decapitation, the brains were rapidly removed and dissected on ice. The rostral part, including the striatum, was processed for Golgi staining (left hemisphere) and HPLC analysis of catecholamines (right hemisphere). The caudal part including the substantia nigra was processed for immunohistochemistry.

### 2.2. Golgi Staining and Analysis

For quantification of morphological changes, the left hemispheres were stained with a Golgi staining kit (FD Rapid Golgi Stain Kit PK-401, FD Neuro Technologies, Columbia, MD, USA) following the manufacturer’s instructions. In brief, tissue was impregnated for 2 weeks before being cut into 170 μm slices using a vibratome (5100mz-135, Campden Instruments, Loughborough, UK). Slices were washed twice with water for 4 min, incubated in the staining solution for 10 min and washed again. Slices were then dehydrated and coverslipped with Entellan (Merck Millipore).

Four slices of each brain located around Bregma level 0.02 mm [26] were selected using the anterior commissure as a landmark and imaged using an inverted microscope (IX81S1F, Olympus, Hamburg, Germany). Type 1 MSN were identified by their spine-free, round or oval soma (12–18 μm) and at least 3–4 branches and their densely spined distal dendrites [27]. Three ventral and three dorsal MSNs were selected per section. For each neuron, images with 20× and 60× objectives (oil immersion, NA 1.35) were acquired as z-stacks with 0.5 μm distance using the software xCellence v2.0 (Olympus, Hamburg, Germany).

Analyses were carried out using ImageJ (FiJi v2.0.0, National Institutes of Health (NIH), Bethesda, Maryland) after minimum intensity projection of the z-stacks. The total dendritic length was measured using the plugin NeuronJ [28]. The number of branch points and the number of spines in a 10 µm stretch of dendrite were determined manually using the multipoint tool of ImageJ. From the total dendritic length (TDL) and the number of branch points (BP) we calculated the segment length (SL) as SL = TDL/BP.

### 2.3. HPLC Analysis of Catecholamines

In order to quantify the loss of dopaminergic axon-terminals in MPTP-treated mice, striatal catecholamine concentrations were measured by HPLC with electrochemical detection at 800 mV. Striatal tissue was homogenized with 50 μL of 0.1 M perchloric acid per mg of tissue using ceramic beads and a 30 s pulse of a speed mill (P12, Analytik Jena AG, Jena, German). Cell debris was removed by centrifugation (17,000× *g*, 20 min at 4 °C) and 20 μL of sample or standard was injected onto the reverse phase column (Prontosil 120-3-C18, Thermo Fisher, Waltham, MA, USA) and kept at 25 °C. The mobile phase consisted of 85 mM sodium acetate, 35 mM citric acid, 0.5 mM octane sulfonic acid, 0.15 mM EDTA, and 10% methanol (pH 4.3). Flow rate was 0.8 mL/min. Retention times were ≈5.5 min for DA, ≈4.7 min for DOPAC and ≈10.9 min for HVA. Peaks were identified manually using Chromeleon software (v6.80, Thermo Fisher, Waltham, MA, USA) and compared to external standards run every 3 samples.

### 2.4. Immunohistochemistry and Quantification

The posterior parts of the brain, including the SN, were fixed for 24 h in 4% paraformaldehyde (pH 7.4), cryoprotected in 30% sucrose for 2 days at 4 °C, frozen by immersion in isopentane (−45 °C) and then stored at −80 °C until serially cut into 30 µm coronal sections. Every third section spanning the SN was stained for tyrosine hydroxylase (TH). The free-floating brain sections were washed three times in Tris-buffered saline (TBS) with 0.1% TritonX (TBS-T). Endogenous peroxidase was blocked by incubation with 0.3% H_2_O_2_ in TBS-T for 30 min followed by three washing steps with TBS-T. The primary anti-TH antibody (rabbit polyclonal, Merck Millipore) was incubated overnight at 4 °C in a dilution of 1:1000 in TBS-T containing 3% normal goat serum (Vector Laboratories). The sections were washed again with TBS-T and incubated with the secondary antibody (biotinylated goat anti-rabbit IgG, Vector Laboratories) in a dilution of 1:200 in TBS-T for 30 min. Subsequently, the sections were washed in PBS, incubated with Avidin-Biotin Complex (Thermo Fisher) for 30 min, followed by another washing step. Visualization was performed via diaminobenzidine (DAB, Vectastain^®^ABC-Kit Standard PK-4000, Vector Laboratories, Burlingame, CA, USA) in a dilution of 1:20 in PBS for 10 min. Sections were mounted on microscope slides after dehydration sections and coverslipped with Entellan (Merck Millipore, Burlington, MA, USA).

TH-positive (dopaminergic) cells in the lateral SNc of the right hemisphere were stereologically counted using the optical fractionator method (StereoInvestigator v11, MicroBrightField, Williston, VM, USA) as described previously [29]. In brief, neurons were manually identified in 50 × 50 µm counting frames presented by the software using an Axioskop 2 microscope (Carl Zeiss Vision) and an oil immersion 63× objective (NA 1.4). Grid size was 50 × 50 µm, and every third section was analyzed. Counts were performed blinded for treatment and timepoint.

### 2.5. Statistical Analysis and Data Visualization

Statistical analyses were performed using GraphPad Prism 5.0 (GraphPad Software, San Diego, CA, USA). The animal-based values—i.e., the number of TH-positive neurons in the substantia nigra, striatal dopamine concentration and striatal metabolite ratio, were normally distributed (Pearson Omnibus normality test). In the graphs they are represented as mean ± SEM and in addition as markers for each animal. Groups were compared by unpaired t-test or two-way ANOVA as indicated in the text.

The values of TDL, branch points, segment length and spine density for each MSN were not normally distributed in all groups. In the graphs they are therefore represented by box and whisker diagrams and groups were compared by the nonparametric Mann–Whitney test. The number of animals and MSN included in the analyses are listed in Table 1.

The entire dataset was analyzed by fitting linear models using the function “lm” and comparing their performance using the function “anova” in RStudio 1.3 following a tutorial by Achim Zeileis (Wirtschaftsuniversität Wien). In order to determine whether a factor (e.g., treatment) had a significant effect (e.g., on segment length SL), we performed the trivial fit f1 < −lm(SL ~ 1, data = d) and the fit for treatment fT < −lm(SL ~ Treatment, data = d) and then compared the results using anova(f1, fT). In order to determine whether a second factor, e.g., the striatal dopamine concentration DA, contains significant additional information, we performed the fit for both factors fDT < −lm(SL ~ DA + Treatment, data = d) and then compared the results using anova(f1, fT, fDT). The different models and the *p* values resulting from the anova analyses are listed in Table 2.

In addition, we performed principal component analysis (PCA) using Rstudio 1.3 and the function PCA of the FactoMineR package. Factors were displayed using the function corrplot of the corrplot package.

## 3. Results

### 3.1. Reduced Complexity of MSN Dendritic Arbors after Dopamine Depletion

In order to deplete dopaminergic neurons of the SN in mice, we used the subacute regimen of MPTP where mice are injected i.p. with 30 mg/kg of MPTP on five consecutive days (Figure 1A). Neurodegeneration is usually completed 14 days after the last MPTP injection. In order to allow sufficient time for the morphological changes in MSN to occur we analyzed the brains 21 days after the last MPTP injection. This interval was chosen based on the time window used in the 6-OHDA model [14,21,23]. At this point, 37% of dopaminergic (TH-positive) neurons were lost in the lateral SN (Figure 1B left column, quantified in C). The concentration of striatal dopamine was reduced to about 34% of NaCl-treated animals (Figure 1E). The relative amount of dopamine metabolites was increased (Figure 1F), indicative of increased dopamine turnover.

MSN were identified in Golgi-stained coronal sections of the striatum based on their typical morphology. Total dendritic length (TDL), the number of dendritic branch points (BP) and the segment length SL = TDL/BP were determined (Figure 2A). On average, TDL was 15% shorter in MSN from MPTP-treated animals than in controls (Figure 2B, median 823 µm in NaCl and 698 µm in MPTP, *p* = 0.0003, Mann–Whitney test). BP were on average 26% lower (Figure 2C, median 9 in NaCl and 6 in MPTP, *p* < 0.0001, Mann–Whitney test). SL was on average 22% longer (Figure 2D, median 90 µm in NaCl and 104 µm in MPTP, *p* = 0.0018, Mann–Whitney test), confirming that the dendritic tree of MSN not only gets smaller but also becomes less complex.

TDL and BP are displayed on a per-animal basis in Appendix A. The dataset was subsequently analyzed by a linear model. Consistent with the nonparametric analysis cited above, treatment (MPTP vs. NaCl) was a significant predictor for TDL, BP and SL (Table 2, #1–3). Striatal dopamine concentration was also a significant predictor for TDL, BP and SL (Table 2, #4–6), but did not provide additional information beyond the treatment category (Table 2, #1–3). Next, we searched for regional differences in the response of MSN to dopamine depletion, but neither a rostral vs. caudal location nor a dorsal vs. ventral location was a significant factor for TDL, BP or SL, and neither location affected their response to MPTP treatment (not shown). We then were interested to know whether BP and TDL generally change together or whether one of them responds more readily. BP was a significant factor for TDL and vice versa (Table 2, #7,8), indicating that they generally change together. Yet, adding the treatment information after TDL improved prediction of BP whereas adding the treatment information after BP did not improve prediction of TDL (Table 2, #7,8). This indicates that the values for BP already contain most of the treatment information and that BP is a more sensitive reporter of MSN structural changes. Accordingly, prediction of TDL was improved by adding the BP values on top of the treatment information (Table 2, #9). The predominant change of BP as compared to TDL in response to dopamine depletion is unlikely a purely statistical effect because the CV of TDL is actually lower than for BP (29% as compared to 41% in the NaCl group). Rather, this finding indicates that MSN predominantly loses short distal branches in response to dopamine depletion, which reduces BP more strongly than TDL, and this is also more plausible to envisage from a biological point of view.

### 3.2. Reduced Spine Density after Dopamine Depletion

The density of MSN dendritic spines was analyzed in high magnification images of Golgi-stained striatal sections on stretches of distal dendrites that were roughly 10 µm long and largely remained in one focal plane (Figure 3A). On average, spine density was 9% lower in MSN from MPTP-treated animals than in controls (Figure 3B, median 9/10 µm in NaCl and 8/10 µm in MPTP, *p* = 0.0018, Mann–Whitney test). Spine density is also displayed in Appendix A.

In the linear model, the spine density was predicted by treatment and dopamine concentration (Table 2, #10,11). As for TDL, BP and SL (Table 2, #1–6), the dopamine concentration did not provide more information than the treatment and there were no regional differences between MSN (not shown). Because spine density behaved differently from TDL in previous studies, we were curious to see whether their changes occurred in the same cells. To this end we performed a principal component analysis with the factors depicted in Figure 3C. The strong contribution of the striatal catecholamines to the first dimension is consistent with the fact that catecholamines respond most strongly to dopamine depletion by MPTP treatment. The stronger contribution of BP as compared to TDL is consistent with the observations from the linear model described above. Interestingly, TDL and BP are the main contributors to the second dimension whereas spine density and SL are the main contributors to the third dimension. The fact that spine density and TDL are grouped in separate dimensions by the PCA is consistent with their differential behavior in previous studies and with the hypothesis that they are regulated by different signaling pathways.

In the linear model, spine density was predicted by TDL and BP (Table 2, #12,13), consistent with the fact that all three are affected by dopamine depletion. TDL predicted spine density better than BP because model fit was improved by adding TDL after BP but not by adding BP after TDL (Table 2, #12,13). Yet, even the model for spine density that already included TDL and BP was improved by adding the treatment information (Table 2, #14), indicating that some MSN change spine density in response to MPTP that do not change TDL or BP. This is again consistent with the notion that the regulation of spine density is not identical with the regulation of dendritic changes (TDL and BP).

### 3.3. Recovery of MSN Dendritic Arborization with Axonal Sprouting

In order to determine whether these changes resulting from MPTP-induced dopamine depletion are reversible when nigrostriatal axon terminals recover, we analyzed MSN morphology 90 days after the last MPTP injection (Figure 1A). In this cohort, the number of dopaminergic neurons was reduced to a similar extent as at 21 days (Figure 1B, right column, quantified in 1D). As expected, the concentration of striatal dopamine was higher at 90 days than at 21 days (Figure 1E). Recovery of the dopamine concentration was incomplete, but dopamine metabolites at 90 days were similar as in NaCl-treated controls (Figure 1F), suggesting that dopamine turnover has returned to normal at this time point.

Morphology of striatal MSN was analyzed in the same way as described above (Figure 4A). TDL was not significantly different between MPTP-treated animals and NaCl-treated controls (Figure 4B, median 1072 µm in NaCl and 1056 µm in MPTP, *p* = 0.1046, Mann–Whitney test), suggesting that TDL was able to recover between 21 days and 90 days. Similarly, the number of branch points was not significantly different between MPTP-treated animals and controls at 90 days (Figure 4C, median 10 in both groups, *p* = 0.6069, Mann–Whitney test). 

The segment length was significantly shorter in the MPTP treated cohort as compared to controls (Figure 4D, median 106 and 116 µm, *p* = 0.0191, Mann–Whitney test), which is the opposite difference to that observed in the 21 days cohort (Figure 2D). Similarly, we observed a statistically higher spine density in MSN analyzed 90 days after MPTP as compared to controls (Figure 5A,B, median 8/10 µm in NaCl and 9/10 µm in MPTP, *p* = 0.0029, Mann–Whitney test). The density of MSN spines in the MPTP group at 90 days was significantly higher than in the MPTP group at 21 days (median 9/10 µm vs. 8/10 µm, *p* = 0.0281, Mann–Whitney test), confirming that spine density increased with the recovery of dopaminergic axon terminals. The difference between the MPTP and NaCl groups at 90 days hence mainly results from the fact that in the NaCl group, the spine density at 90 days was smaller than at 21 days (median 9/10 µm at 21 days and 8/10 µm at 90 days, *p* = 0.0009, Mann–Whitney test).

In the linear model analysis of the 90 days cohort, dopamine and treatment did not predict TDL or the number of branch points or segment length (not shown), consistent with the lacking difference between MPTP and NaCl treatment (Figure 4B–D). Spine density was predicted by treatment and dopamine, consistent with the observed difference (Figure 5B). Yet, dopamine concentration was the stronger predictor because it added significant information when added after treatment whereas before treatment it did not (Table 3 #1,2).

When we analyzed the MPTP group only, spine density was predicted by the number of TH neurons in the substantia nigra and by the dopamine concentration in the striatum (Table 3, #3,4, first predictors), indicating that spine density is influenced both by the extent of MPTP-induced degeneration of TH-positive neurons in the substantia nigra and by the recovery of dopaminergic axon terminals in the striatum. Dopamine was the stronger predictor because it remained a significant factor after TH whereas TH was not (Table 3, #3,4, second predictors). TDL, in contrast, was predicted more strongly by the number of TH-positive neurons than by striatal dopamine (Table 3, #5,6). This is consistent with the hypothesis that dendritic arborization and spine density are independently regulated and with the hypothesis that spine density recovers more readily after dopamine depletion than dendritic arborization.

## 4. Discussion

In this study we observed morphological alterations in striatal MSN after MPTP-induced degeneration of dopaminergic neurons that are consistent with findings in PD patients and other animal models. Following the recovery of dopaminergic axon terminals, these morphological changes were reversible. Reversibility in spine density has been observed in previous studies with exogenous administration of levodopa. Yet, this is, to our knowledge, the first study showing reversibility of dendritic arborization as reported by TDL and branch points.

To analyze MSN morphology, we used the classical Golgi staining that was also used in studies on PD patients [19,20,30] and in several of the earlier studies on animal models [9,13]. The Golgi staining allows reconstruction of dendritic arbors of individual neurons and quantification of dendritic spines without the need to fill individual neurons by a fluorescent dye. This advantage allowed us to analyze a much larger number of MSN than in studies with individually filled MSN [14,21,22,23]. The major disadvantage of the Golgi method lies in the fact that it cannot be easily combined with immunohistochemistry. Consequently, we were not able to discriminate between D1-MSN and D2-MSN. In previous work by others, notable differences were observed between D1-MSN, D2-MSN, and D1/D2 double positive MSN. In D2-MSN, a reduction in spine density was observed already 2–3 weeks after dopamine depletion whereas in D1-MSN spine density was only reduced with more chronic dopamine depletion [7,14,21,22,31,32].

The density of striatal spines we observed (Figure 3B) was similar as in previous studies in mice [13,14,15,21,23]. The reduction of spine density in response to dopamine depletion we observed after MPTP (Figure 3B) was also observed in mice with 6-OHDA-induced dopamine depletion [14,15,21], in Pitx3-/-mice [22], rats [16,17,18], monkeys [9] and PD patients [20,30]. The magnitude of the reduction was smaller in our data than in previous studies. This could be related to the smaller extent of dopamine depletion in the MPTP model as compared to the 6-OHDA models and PD patients.

When measured at 90 days after MPTP, spine density recovered almost completely (Figure 5B). This is noteworthy given the moderate recovery of dopamine in this cohort (Figure 1E). Yet, we assume that the recovery of nigrostriatal axon terminals was more pronounced than the recovery of dopamine. In previous experiments [24,29], nigrostriatal axon terminals returned almost to baseline levels. This is consistent with the normalized ratio of dopamine metabolites (Figure 1F) and can explain the pronounced recovery of spine density (Figure 5B). In the 6-OHDA model and in Pitx3-/-mice, reversibility of spine density is observed in D2-MSN but not in D1-MSN after 1–2 weeks of levodopa administration [21,22,23]. It will therefore be interesting to determine in future studies whether reversibility of spine density is different between D1-MSN and D2-MSN at 90 days after MPTP. Of note, reversibility of spine density was previously observed in the MPTP model after a treadmill exercise [13]. This finding is reminiscent of a finding from the ELLDOPA study on PD patients where the long-term beneficial effects of levodopa medication were more pronounced in the dominant hand, i.e., in the hand that receives more exercise [33]. It indicates that exercise can to some extent mimic the effects of levodopa administration. 

Our values for TDL and branch points were measured in reconstructions of minimum intensity projections obtained from coronal slices. They were in a similar range as that obtained from maximum intensity projections of MSN filled with lucifer yellow in coronal slices [14]. Both values are 2–3 times shorter than TDL obtained with 3D analysis of two-photon images from parasagittal slices of a similar thickness [21]. This difference indicates that we substantially underestimate the extent of the dendritic tree with 2D projections. The number of dendritic branch points we observed was similar to the 3D analysis [21], confirming the validity of our analysis. Because we are primarily interested in relative changes, the systematic underestimation of TDL is not problematic. It nonetheless illustrates how the dendritic arbor is altered when neurons grow on coverslips or plates instead of the 3D environment of the brain. A much larger dendritic arbor could explain the quite substantial differences recently observed between matrigel-based 3D neuronal cultures as compared to traditional 2D cultures [34,35].

With MPTP-induced dopamine depletion we observed a reduction in TDL (Figure 2B), consistent with previous findings in mice using MPTP [13] or 6-OHDA [14,21], and with findings in monkeys [9] and PD patients [19,20,30]. In contrast to the recovery of TDL we observed in our MPTP model, TDL was not reversible after levodopa administration in the 6-OHDA model [21]. There are several possible explanations that will be interesting to explore in future studies. First, dopamine depletion was more pronounced in the cited 6-OHDA models than in our MPTP model. This might affect the possibility for TDL recovery. Furthermore, levodopa dosing in the cited studies was chosen to induce dyskinesias and has not been optimized for TDL recovery. Finally, secretion of endogenous dopamine in our study might have different effects to exogenous administration of levodopa. Exogenous levodopa is at least partially converted in serotonergic terminals [4] and glutamate is co-released from nigrostriatal terminals [36,37]; such co-transmitters could thus be important for the maintenance and recovery of MSN morphology. In addition, adaptation in cortico-striatal glutamatergic signaling, which is known to affect MSN spine density [38], could underlie the changes in MSN spine density after dopamine depletion and recovery.

Taken together, our findings indicate that all consequences of dopamine depletion on the morphology of striatal MSN can be reversible. Of course, further research is required to mechanistically determine whether MSN changes after degeneration and recovery of dopaminergic axon terminals really result from the changes in dopaminergic neurotransmission or from other factors. Similarly, further work will be required to determine the molecular pathways that govern the adaptation of MSN arborizations and spine density. Additional interventions such as acute or chronic administration of dopamine agonists and antagonists in animals with different extents of dopaminergic denervation will be required. Still, this encouraging finding suggests that it should be possible to design regimens of dopamine substitution for PD patients that avoid structural changes in striatal MSN. Such regiments could potentially also avoid the development of dyskinesias. It is already established that pronounced depletion of striatal dopamine is necessary for levodopa-induced dyskinesias in rodents. If the smaller extent of dopamine depletion in the MPTP model is confirmed as the critical factor for reversibility, these findings indicate that dopamine substitution needs to be started early in the disease course to avoid long-term motor complications.

## Figures and Tables

**Figure 1 cells-09-02441-f001:**
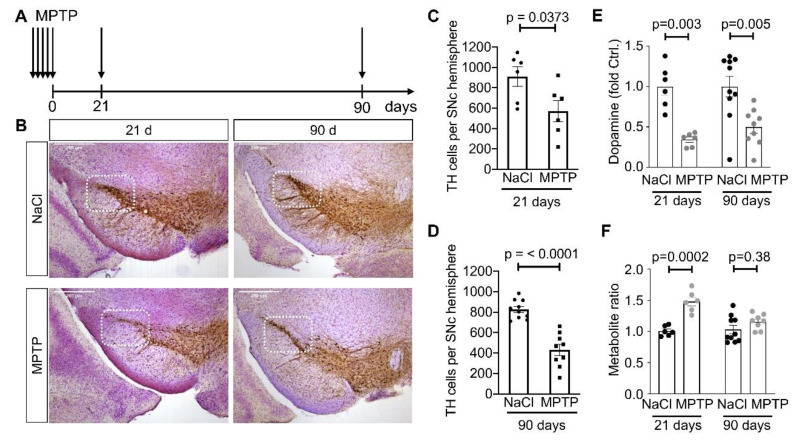
Dopaminergic phenotype in the MPTP model. (**A**) Schematic representation of the MPTP model. Mice received either MPTP or saline only (NaCl) on five consecutive days. They were analyzed either 21 days or 90 days after the last MPTP administration. Scale bars 200 µm. (**B**) Representative images of TH stained coronal midbrain sections of animals 21 days or 90 days after the last MPTP injection. The framed area of the Substantia nigra pars compacta (SNc) represents the lateral area analyzed by the stereological counting of dopaminergic neurons. (**C**) Number of TH-positive cells after 21 days. *p*-value is from unpaired *t*-test. (**D**) Number of TH-positive cells after 90 days. *p*-value is from unpaired t-test. When panels (**C**) and (**D**) were analyzed by two-way ANOVA (Df = 1 for both factors), we observed a significant difference for treatment (F = 30.05) but no significant difference for time point (F = 2.81). (**E**) Striatal dopamine concentration, measured by HPLC, is relative to the mean of the NaCl condition. Two-way ANOVA (Df = 1 for both factors) revealed a significant effect of treatment (F = 25.33) but not time point (F = 0.34). (**F**) Striatal concentration of dopamine metabolites, i.e., (DOPAC+HVA)/dopamine, relative to the mean of the NaCl condition. Two-way ANOVA (Df = 1 for both factors) revealed a significant effect of treatment (F = 31.74) but not time point (F = 2.58). Each marker represents one animal. Numbers of animals are in Table 1.

**Figure 2 cells-09-02441-f002:**
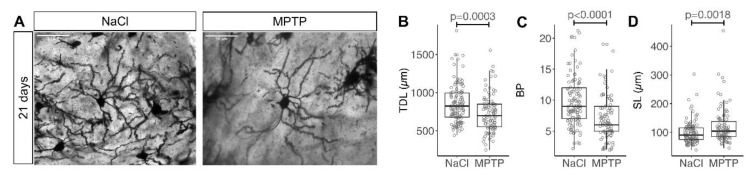
Dendritic arborization of striatal MSN 21 days after MPTP. (**A**) Representative images of Golgi stained MSN in the striatum analyzed 21 days after the last MPTP injection. Scale bars 50 μm. (**B**) Total dendritic length (TDL), (**C**) number of branch points (BP) per MSN, (**D**) segment length (SL = TDL/BP). Each marker represents one MSN. Numbers of MSN are in Table 1. *p*-values are from Mann–Whitney tests.

**Figure 3 cells-09-02441-f003:**
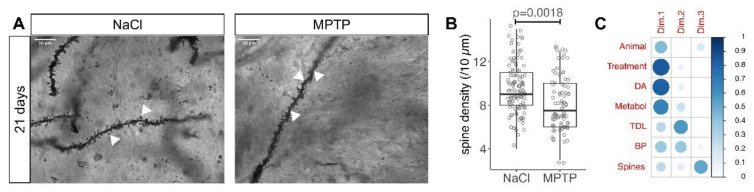
Spine density in MSN 21 days after MPTP. (**A**) Representative images of Golgi-stained MSN dendrites (arrowheads) 21 days after the last MPTP injection. Scale bars 10 μm. (**B**) Number of spines in a 10 µm stretch of dendrite. Each marker represents one MSN. Numbers of MSN are in Table 1. *p*-value is from Mann–Whitney test. (**C**) Illustration of the principal component analysis with size and color of the blue circles representing the importance (cos^2^) of the factor in that row for the dimension in that column.

**Figure 4 cells-09-02441-f004:**
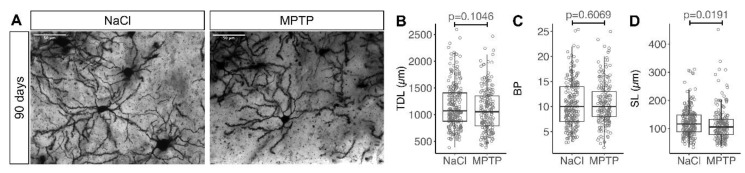
Recovery of dendritic arborization 90 days after MPTP. (**A**) Representative images of Golgi stained MSN in the striatum analyzed 90 days after the last MPTP injection. Scale bars 50 μm. (**B**) Total dendritic length (TDL), (**C**) number of branch points (BP) per MSN, (**D**) segment length (SL = TDL/BP). Each marker represents one MSN. Numbers of MSN are in Table 1. *p* values are from Mann–Whitney tests.

**Figure 5 cells-09-02441-f005:**
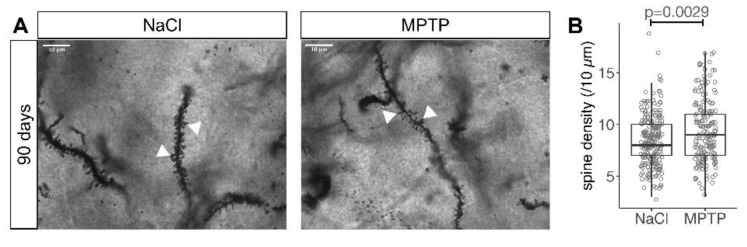
Recovery of spine density 90 days after MPTP. (**A**) Representative images of Golgi stained MSN dendrites (arrowheads) 90 days after the last MPTP injection. Scale bars 10 μm. (**B**) Number of spines in a 10 µm stretch of dendrite. Each marker represents one MSN. Numbers of MSN are in Table 1. *p* value is from Mann–Whitney test.

**Table 1 cells-09-02441-t001:** Number of mice and MSN for morphological analyses.

Treatment	Days after Treatment	Mice ^1^	SN ^2^	HPLC ^3^	Golgi ^4^	MSN ^5^	Spine Density ^6^
NaCl	21	6	6	6	5	118	116
MPTP	21	6	6	6	4	95	85
NaCl	90	10	10	10	10	240	235
MPTP	90	10	9	9	8	192	182

^1^ Number of mice that started the experiment; ^2^ Number of mice/stained SN available for stereology of TH neurons. One animal died during the course of the experiment; ^3^ Number of mice analyzed for HPLC; ^4^ Number of mice/stained striata available for MSN analysis. Three brains were damaged during vibratome sectioning; ^5^ Number of MSN analyzed for TDL and branch points; ^6^ Number of MSN analyzed for spine density. Within one brain, dendritic segments suitable to determine spine density were not found for each cell.

**Table 2 cells-09-02441-t002:** Analyses of MSN morphology at 21 days by linear models.

#	Response	Predictors		*p* Values ^1^	
		First	Second	First	Second
1	TDL	Treatment	DA	3.362e−4	n.s.
2	BP	Treatment	DA	2.258e−5	n.s.
3	SL	Treatment	DA	5.357e−3	n.s.
4	TDL	DA	Treatment	2.711e−3	3.7215e−2
5	BP	DA	Treatment	1.494e−4	6.3311e−3
6	SL	DA	Treatment	9.064e−3	n.s.
7	TDL	BP	Treatment	<2e−16	n.s.
8	BP	TDL	Treatment	<2e−16	4.168e−3
9	TDL	Treatment	BP	1.187e−6	<2e−16
10	spines	Treatment	DA	5.948e−6	n.s.
11	spines	DA	Treatment	8.435e−4	3.245e−4
12	spines	TDL	BP	2.196e−9	n.s.
13	spines	BP	TDL	3.188e−6	6.618e−5
14	spines	BP + TDL	Treatment	4.139e−5	4.852e−4

^1^ The first *p* value reports the comparison of the trivial fit to the fit with the first predictor. It therefore reports whether the first predictor contributes meaningful information. The second *p* value reports the comparison of the model with the first predictor to the model with both predictors. The second *p* value therefore reports whether the second predictor adds meaningful information to the model. Abbreviations: TDL, total dendritic length; BP, number of dendritic branch points; SL, segment length; DA, striatal dopamine concentration; n.s., non-significant.

**Table 3 cells-09-02441-t003:** Analyses of MSN morphology at 90 days by linear models.

#	Group	Response	Predictors		*p* Values	
			First	Second	First	Second
1	all 90d	spines	Treatment	DA	2.891e−4	3.43835e−2
2	all 90d	spines	DA	Treatment	1.223e−4	n.s.
3	MPTP only	spines	TH	DA	3.5920e−3	1.599e−4
4	MPTP only	spines	DA	TH	3.022e−6	n.s.
5	MPTP only	TDL	TH	DA	2.826e−3	1.302e−3
6	MPTP only	TDL	DA	TH	n.s.	3.174e−5

Abbreviations: TDL, total dendritic length; DA, striatal dopamine concentration; TH, number of TH-positive neurons in substantia nigra; n.s., non-significant.

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
