# Peer review of "Changes in Striatal Medium Spiny Neuron Morphology Resulting from Dopamine Depletion Are Reversible"

_cells, 2020, doi:10.3390/cells9112441_

Round 1

Reviewer 1 Report

witzig et al. contribute an interesting study about the reversal of striatal MPTP-induced morphological changes in a MPTP mouse model of nigrostriatal degeneration. This well-conducted, although low powered, the study adds to the knowledge about the possible role of DA denervation upon striatal morphological changes that might then be responsible for both occurrence of abnormal involuntary movements (i.e. dyskinesias) and maladaptive response to antiparkinsonian therapies. This reviewer's enthusiasm would be enhanced if the study had investigated the causative role of DA content instead of conducting another correlative study (through the use of DA-/- mice, TH-CRISP-Cas9 mice, ICV DA infusion in fully depleted animals, etc...

Such limitations should be acknowledged and discussed.

Why not compare with two-way ANOVAs the loss of TH+ neurons? 

F values and degrees of freedom to be provided for all ANOVAs.

Author Response

Witzig et al. contribute an interesting study about the reversal of striatal MPTP-induced morphological changes in a MPTP mouse model of nigrostriatal degeneration. This well-conducted, although low powered, the study adds to the knowledge about the possible role of DA denervation upon striatal morphological changes that might then be responsible for both occurrence of abnormal involuntary movements (i.e. dyskinesias) and maladaptive response to antiparkinsonian therapies. This reviewer's enthusiasm would be enhanced if the study had investigated the causative role of DA content instead of conducting another correlative study (through the use of DA-/- mice, TH- CRISP-Cas9 mice, ICV DA infusion in fully depleted animals, etc... Such limitations should be acknowledged and discussed.

-> We thank the reviewer for suggesting these follow-up experiments, which are of course necessary to better understand the process of adaptation to both dopamine depletion and substitution. They will be subject of further studies, but are beyond the scope of the present one.

-> We have expanded the section on limitations of this study on page 17. “Of course, further research is required to mechanistically determine whether MSN changes after degeneration and recovery of dopaminergic axon terminals really result from the changes in dopaminergic neurotransmission or from other factors. Similarly, further work will be required to determine the molecular pathways that govern adaptation of MSN arborisations and spine density. Additional interventions such as acute or chronic administration of dopamine agonists and antagonists in animals with different extents of dopaminergic denervation will be required.”

Why not compare with two-way ANOVAs the loss of TH+ neurons?

-> We had analyzed the cohorts separately because they were acquired separately. But we followed the advice of the reviewer and re-analyzed the cohorts by two-way ANOVA.

F values and degrees of freedom to be provided for all ANOVAs.

-> F values and Df were added to the legend of Figure 1.

Reviewer 2 Report

This is an interesting paper, that revisits the concept of reversible changes in spine density of medium spiny neurons (MSNs) after dopamine depletion and subsequent restoration (seen for example after L-dopa administration). In this case, the authors use the MPTP model of nigrostriatal degeneration in mice. The work confirms recovery of spine density long-term after neurodegenerative treatment (observed in other PD models), and adds data about dendritic arborization, currently lacking in the field. The authors make use of a classical but effective tool, such as Golgi staining, and combine it with modern image analysis, which allows them to analyse a very large amount of neurons (a strength of the paper). The experimental design is clear and statistical analysis are explained in detail, which is welcomed. The text is clear and the figures well presented. However, I believe that the authors could extract further results from the data they have, and some of the observations could be explained by other means:

Major comments

  1. The authors represent TH cell counts and HPLC results as per animal (i.e n=6 or 10). However, the data from spine morphology analysis are expressed as per neuron (n > 150). I believe the study would benefit if all morphology results were expressed also as per animal (in addition to the current representation), because:

  • The variability between animals is high, as typical for in vivo experiments. This can be seen both in the TH and Dopamine quantifications. This variability per animal should be reflected (or at least investigated) in morphology as well.
  • (even more important than point “a”) A per animal analysis would allow the authors to perform a correlation analysis, in order to determine if the level of neurodegeneration in each animal corresponds to the changes observed in morphology. This would strongly support the main hypothesis of the paper.

  1. The authors claim that at 3 months post MPTP treatment, dopaminergic terminals are restored in the striatum via axonal sprouting (they cite their PNAS paper from 2007), and they state in the discussion that “following recovery of striatal dopamine…” (line 350). However, the data from figure 1E (misnamed as D in the figure, please correct) does not support this statement, since dopamine levels are not restored after 90 days. The authors should at least discuss this, given that their hypothesis of reversible changes in MSN morphology is based on this statement.

  1. Decrease in spine density in MSNs can be attributed also to excess of corticostriatal glutamatergic signal, driven by the absence of dopamine-mediated modulation (See Garcia et al., 2010, Cerebral Cortex). This study also suggests that the striatal spine loss as a consequence of nigral degeneration is reversible. Perhaps the reversal of morphological changes the authors observe in MSN despite absence of dopamine (which levels are only slightly restored at 90 days) are due to compensatory mechanisms that autoregulate cortico-striatal glutamatergic tone.

Minor comments

-Line 108, typo: HPLC.

-Table 1: only 5 of 6 footnotes are described. Also, descriptions for 4 and 5 should be for 5 and 6.

-Line 207, missing word: days

-Figure 1, there are two panels D.

-Figures: scale bars are too small

-Line 393, typo: range

-The table 2, depicting the results from the model, is confusing. The data is there, but I wonder if there is not a visual representation that might facilitate rapid visualization of which predictor is best.

-Perhaps comparison between 21 and 90 days for morphology would make more sense if the data were together in the same figure (Fig 2 and 4, and 3 and 5). But, as the previous comment, this is just a matter of taste.

Author Response

This is an interesting paper, that revisits the concept of reversible changes in spine density of medium spiny neurons (MSNs) after dopamine depletion and subsequent restoration (seen for example after L-dopa administration). In this case, the authors use the MPTP model of nigrostriatal degeneration in mice. The work confirms recovery of spine density long-term after neurodegenerative treatment (observed in other PD models), and adds data about dendritic arborization, currently lacking in the field. The authors make use of a classical but effective tool, such as Golgi staining, and combine it with modern image analysis, which allows them to analyze a very large amount of neurons (a strength of the paper). The experimental design is clear and statistical analysis are explained in detail, which is welcomed. The text is clear and the figures well presented. However, I believe that the authors could extract further results from the data they have, and some of the observations could be explained by other means:

Major comments

  1. The authors represent TH cell counts and HPLC results as per animal (i.e n=6 or 10). However, the data from spine morphology analysis are expressed as per neuron (n > 150). I believe the study would benefit if all morphology results were expressed also as per animal (in addition to the current representation), because:

The variability between animals is high, as typical for in vivo experiments. This can be seen both in the TH and Dopamine quantifications. This variability per animal should be reflected (or at least investigated) in morphology as well.

(even more important than point “a”) A per animal analysis would allow the authors to perform a correlation analysis, in order to determine if the level of neurodegeneration in each animal corresponds to the changes observed in morphology. This would strongly support the main hypothesis of the paper.

-> The results are now also represented as per animal in the new supplemental Figure S1, which includes the suggested correlation with dopamine levels. Figure S1 is referred to on pages 11 and 12.

  1. The authors claim that at 3 months post MPTP treatment, dopaminergic terminals are restored in the striatum via axonal sprouting (they cite their PNAS paper from 2007), and they state in the discussion that “following recovery of striatal dopamine…” (line 350). However, the data from figure 1E (misnamed as D in the figure, please correct) does not support this statement, since dopamine levels are not restored after 90 days. The authors should at least discuss this, given that their hypothesis of reversible changes in MSN morphology is based on this statement.

-> As discussed on page 16, the extent to which the dopamine concentration is restored is lower as the extent to which the density of striatal axon terminals is restored. In order to acknowledge this point, we have replaced in the text “dopamine concentration” by “recovery of dopaminergic axon terminals”. In addition, we argue that the recovery of striatal morphology and the normalization of the striatal catecholamine metabolite ratio indicate that recovery of striatal catecholamines is sufficient.

  1. Decrease in spine density in MSNs can be attributed also to excess of corticostriatal glutamatergic signal, driven by the absence of dopamine-mediated modulation (See Garcia et al., 2010, Cerebral Cortex). This study also suggests that the striatal spine loss as a consequence of nigral degeneration is reversible. Perhaps the reversal of morphological changes the authors observe in MSN despite absence of dopamine (which levels are only slightly restored at 90 days) are due to compensatory mechanisms that autoregulate cortico-striatal glutamatergic tone.

-> Thank you for this suggestion, which has not been in the focus of our “dopamine-centric” view. We have added it on page 17.

Minor comments

-Line 108, typo: HPLC.

-Table 1: only 5 of 6 footnotes are described. Also, descriptions for 4 and 5 should be for 5 and 6.

-Line 207, missing word: days

-Figure 1, there are two panels D.

-Figures: scale bars are too small

-Line 393, typo: range

-The table 2, depicting the results from the model, is confusing. The data is there, but I wonder if there is not a visual representation that might facilitate rapid visualization of which predictor is best.

-Perhaps comparison between 21 and 90 days for morphology would make more sense if the data were together in the same figure (Fig 2 and 4, and 3 and 5). But, as the previous comment, this is just a matter of taste.

-> Thank you for these comments. We have corrected all the typos.

As for Table 2, we have not found a better way to display the data but hope that the new visual representation of the data in supplemental Figure S1 will help illustrate the analyses we did.

As for the combination of the 21 and 90 days data, we have chosen to leave it in the current flow.